# Efficient *ab initio* many-body calculations based on sparse modeling of Matsubara Green's function

Hiroshi Shinaoka[1][*], Naoya Chikano[1], Emanuel Gull[2], Jia Li[2], Takuya Nomoto[7], Junya Otsuki[4], Markus Wallerberger[5], Tianchun Wang[3] and Kazuyoshi Yoshimi[6]

**1** Department of Physics, Saitama University, Saitama 338-8570, Japan
**2** University of Michigan, Ann Arbor, Michigan 48109, USA
**3** Department of Applied Physics, University of Tokyo, Japan
**4** Research Institute for Interdisciplinary Science, Okayama University, Okayama 700-8530, Japan
**5** Institute of Solid State Physics, TU Wien, 1040 Vienna, Austria
**6** Institute for Solid State Physics, University of Tokyo, Chiba 277-8581, Japan
**7** Research Center for Advanced Science and Technology, University of Tokyo, Tokyo 153-8904, Japan

★ h.shinaoka@gmail.com

## Abstract

This lecture note reviews recently proposed sparse-modeling approaches for efficient *ab initio* many-body calculations based on the data compression of Green's functions. The sparse-modeling techniques are based on a compact orthogonal basis, an intermediate representation (IR) basis, for imaginary-time and Matsubara Green's functions. A sparse sampling method based on the IR basis enables solving diagrammatic equations efficiently. We describe the basic properties of the IR basis, the sparse sampling method and its applications to *ab initio* calculations based on the *GW* approximation and the Migdal–Eliashberg theory. We also describe a numerical library for the IR basis and the sparse sampling method, `sparse-ir`, and provide its sample codes. This lecture note follows the Japanese review article [H. Shinaoka *et al.*, Solid State Physics 56(6), 301 (2021)].

# 1   Introduction

Perturbation and quantum field theories based on Green's functions are widely used for *ab initio* and quantum many-body calculations. The imaginary-time formalism based on Matsubara Green's functions is well known for its simplicity in numerical treatment. It has been used for *ab initio* calculations and model calculations based on various theories such as the *GW* approximation [1], the random phase approximation (RPA) [2], the fluctuation exchange approximation (FLEX) [3, 4], the dynamical mean-field theory (DMFT) [5], and quantum Monte Carlo methods [6]. Most readers of this Lecture Note may have had experience using such computational methods. Applications of imaginary-time formalism are not limited to condensed matter physics but is also used in quantum chemistry and high energy physics [7–9].

Although the Matsubara Green's function is used in a wide range of fields, its efficient numerical handling has yet to be fully established. For example, in *ab initio* calculations, it is necessary to simultaneously deal with multiple energy scales that differ by orders of magnitude. Typical energy scales are the width of low-energy bands (several eV to several tens of eV) and the low temperature at which physical phenomena occur ($1 \text{ K} \simeq 0.1 \text{ meV}$). In such cases, the number of imaginary (Matsubara) frequencies required in the calculations increases, and the computation time and the amount of memory required increase to an unmanageable level.

Furthermore, calculations at the two-particle level (e.g., susceptibility calculations using the Bethe-Salpeter equation) require the handling of vertex functions that depend on multiple imaginary frequencies. Therefore, calculations with full frequency dependence are extremely costly even for simple model calculations, and applications to realistic materials are impractical. In this Lecture Note, we review efficient many-body and *ab initio* calculation methods based on the data compression of Matsubara Green's functions. A compact basis for Matsubara

Green's function named intermediate representation (IR) was proposed in 2017 [10, 11]. In 2020, an efficient calculation method based on sparse sampling was developed [12]. Since then, its applications to *ab initio* calculations have been rapidly spreading [13–18].

This Lecture Note summarizes the theoretical progress made during the last few years. A comprehensive overview is given of 1) the basic properties of the IR basis, 2) the sparse sampling method for fast computation of diagrammatic equations, and 3) numerical library `sparse-ir`. We hope that this Lecture Note help readers start using the IR basis and the sparse sampling method in various fields.

## 2 Intermediate-representation (IR) of Matsubara Green's functions

The intermediate representation (IR) is a basis set in which the imaginary frequency and time dependence of the two-point imaginary-time/-frequency correlation function can be expanded compactly [10, 19]. Let us summarize its definition and properties.

The general definition of the imaginary-time two-point correlation function is:

$$G(\tau) = -\langle T_\tau A^\alpha(\tau) B^\alpha(0) \rangle, \tag{1}$$

where $\beta = 1/T$ is the inverse temperature, $\tau$ is the imaginary time, $T_\tau$ is the time ordering operator, and $\alpha$ is either F (fermion) or B (boson). Here, we take $k_B = 1$. In addition, $A$ and $B$ are the operators. In the important special case of a one-particle Green's function, $A$ is an annihilation operator, and $B$ is a creation operator.

The spectral representation of the two-point correlation function is given as follows:

$$G(\tau) = -\int_{-\omega_{\max}}^{\omega_{\max}} d\omega' K^\alpha(\tau, \omega') A(\omega'), \tag{2}$$

where $A(\omega)$ is a corresponding spectral function and the kernel $K^\alpha(\tau, \omega)$ is defined as

$$K^\alpha(\tau, \omega) = \begin{cases} \dfrac{e^{-\tau\omega}}{1 + e^{-\beta\omega}} & (\alpha = F), \\ \dfrac{e^{-\tau\omega}}{1 - e^{-\beta\omega}} & (\alpha = B), \end{cases} \tag{3}$$

for $0 < \tau < \beta$. We take $\omega_{\max}$ to be sufficiently large such that the spectral function $A(\omega)$ is nonzero only for $\omega \in [-\omega_{\max}, \omega_{\max}]$. We can regard Eq. (2) as an integral equation which connects the spectral function $A(\omega)$ to the correlation function $G(\tau)$ through the kernel $K^\alpha(\tau, \omega)$.

To avoid the divergence of the bosonic kernel at $\omega = 0$, we reformulate Eq. (2) as [20]

$$G(\tau) = -\int_{-\omega_{\max}}^{\omega_{\max}} d\omega' K(\tau, \omega') \rho(\omega'), \tag{4}$$

where we call $K(\tau, \omega)[= K^F(\tau, \omega)]$ a *logistic* kernel, and $\rho(\omega)$ is the modified spectral function defined by

$$\rho(\omega) \equiv \begin{cases} A(\omega) & (\alpha = F), \\ \dfrac{A(\omega)}{\tanh(\beta\omega/2)} & (\alpha = B). \end{cases} \tag{5}$$

This allows to use the same kernel for fermions and bosons. Note that a different regularization was used in the original proposal of IR [10].

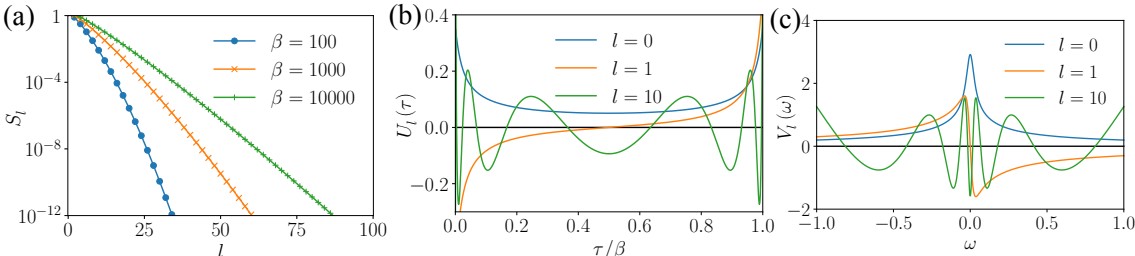

Figure 1: (a) Singular value $S_l$ computed for various values of $\beta$. (b), (c) IR basis functions $U_l(\tau)$ and $V_l(\omega)$ computed for $l = 0, 1, 10$ and $\beta = 100$, respectively. $\omega_{\max}$ is fixed at $\omega_{\max} = 1$.

The task at hand is to find a compact representation of the *imaginary-frequency* (Matsubara) Green's function $G(i\omega)$ that retains full information of the *real-frequency* spectral function $\rho(\omega)$. For this, we perform a singular value expansion of the logistic kernel $K$ for a given $\beta$ and $\omega_{\max}$ [10]:

$$K(\tau, \omega') = \sum_{l=0}^{\infty} U_l(\tau) S_l V_l(\omega') . \tag{6}$$

The Fourier transform of $K(\tau, \omega')$ with respect to $\tau$ yields the singular value expansion in the imaginary-frequency domain

$$\hat{K}^{\alpha}(i\omega^{(\alpha)}, \omega') \equiv -\int_0^{\beta} d\tau K(\tau, \omega') e^{i\omega^{(\alpha)}\tau} = -\sum_{l=0}^{\infty} \hat{U}_l^{\alpha}(i\omega^{(\alpha)}) S_l V_l(\omega'), \tag{7}$$

where

$$\hat{U}_l^{\alpha}(i\omega^{(\alpha)}) \equiv \int_0^{\beta} d\tau U_l(\tau) e^{i\omega^{(\alpha)}\tau}. \tag{8}$$

Here, $i\omega^{(\alpha)}$ is the imaginary frequency corresponding to the statistics $\alpha$ and we attached a hat ˆ to the quantities defined in the imaginary-frequency domain. The minus sign in Eq. (7) originates from the convention $K(\tau, \omega) > 0$. In the following, we will denote $i\omega^{(\alpha)}$ as $i\omega$ for simplicity.

The singular value expansion has similar properties as the singular value decomposition of matrices: The singular values $S_l$ satisfy $S_0 > S_1 > \ldots > 0$. The left singular functions $\{U_0(\tau), U_1(\tau), \ldots\}$ form an orthonormal set on the imaginary time axis $\tau \in [0, \beta]$, while the right singular functions $\{V_0(\omega), V_1(\omega), \ldots\}$ form an orthonormal set on the real frequency axis $\omega \in [-\omega_{\max}, \omega_{\max}]$.

The so-called IR basis functions are nothing but the left singular functions $U_l(\tau)$ $[\hat{U}_l^{\alpha}(i\omega)]$ and the right singular functions $V_l(\omega)$. It should be emphasized that the IR basis functions only depend on the inverse temperature $\beta$, statistics $\alpha$, and the energy (frequency) cutoff $\omega_{\max}$ and do not depend on the details of the system.

Figure 1 shows the singular values $S_l$ and the IR basis functions $U_l(\tau)$ and $V_l(\omega)$. The IR basis functions have the following interesting properties:

**Property 1** The singular values $S_l$ are non-degenerate, non-negative, and monotonically decrease faster than exponentially with respect to $l$ [1].

---

[1] For numerical evidence, refer to Appendix D of [21].

**Property 2** The number of numerically significant singular values (e.g., $S_l/S_0 \geq 10^{-15}$) is determined by the dimensionless quantity $\Lambda \equiv \beta\omega_{\max}$ and increases only logarithmically with respect to $\Lambda$ [21].

**Property 3** $U_l(\tau)$ and $V_l(\omega)$ can be chosen to be real functions, and they then become even (odd) functions for even (odd) $l$. In this convention, $\hat{U}_l^\alpha(i\omega)$ is purely imaginary or real.

**Property 4** $U_l(\tau)$ and $V_l(\omega)$ have $l$ roots. In addition, in the limit of $\beta\omega_{\max} \to 0$ (the high-temperature limit), $U_l(\tau(x))$ and $V_l(\omega(x))$ coincide with the Legendre polynomial $P_l(x)$ up to a constant $[\tau(x) \equiv \beta(x+1)/2, \omega(x) \equiv x\omega_{\max}$ for $-1 < x < 1]$ [2].

We sketch mathematical proofs of Properties 1, 3, and 4 in Appendix A. Property 2 has been verified only numerically[3].

Now, let us expand an "arbitrary" two-point correlation using a finite number ($L$) of the IR basis functions:

$$\hat{G}(i\omega) = \sum_{l=0}^{L-1} \hat{U}_l^\alpha(i\omega)G_l + \epsilon_L, \tag{9}$$

where $\epsilon_L$ is a truncation error. If the corresponding spectral function $\rho(\omega)$ is finite only in the interval $[-\omega_{\max}, \omega_{\max}]$, the expansion coefficient $G_l$ can be evaluated using the orthonormality of the IR functions as

$$G_l = -S_l \int_{-\omega_{\max}}^{\omega_{\max}} d\omega\rho(\omega)V_l(\omega) \equiv -S_l\rho_l. \tag{10}$$

From Eq. (10), it can be seen that $G_l$ vanishes at least as fast as $S_l$, i.e., faster than the exponential function from Property 1.

Figure 2 shows the convergence of the expansion coefficients $\rho_l$ and $G_l$ for some simple spectral functions $\rho(\omega)$ [4] The convergence of $\rho_l$ strongly depends on the shape of $\rho(\omega)$. In general, $\rho_l$ decays exponentially in the case of a smooth spectral function, but does not decay at all in the case of a $\delta$ function spectrum. By contrast, $G_l$ always decays as fast as or even faster than the singular values, regardless of the spectral functions. We would like to emphasize again that this convergence is determined by the singular values of the kernel and does not depend on the details of the system.

In Eq. (10), we can see that when we transform $\rho_l$ into $G_l$, the contribution of $\rho_l$ is suppressed by the singular values exponentially at a large $l$. This means that some of the information in the spectral function is lost in the transformation to the imaginary-time domain. Thus, it is usually difficult to recover the exact spectral function from the numerical data of the Matsubara Green's function. This is because, in the inverse transformation $\rho_l = -(S_l)^{-1}G_l$, the error of $G_l$ is amplified by a small singular value at a large $l$. Many approximate methods based on different principles have been proposed to mitigate this problem, such as the maximum entropy method [23] and the sparse modeling approach (SpM) [11,19]. However, we will not discuss this topic here.

The main focus of this article is to take the advantage of the fact that the Matsubara Green's function has less information than its real-frequency counterpart, and to speed up the computation within the imaginary-time/-frequency domain. In this respect, the IR basis $\hat{U}_l^\alpha(i\omega)$ and $U_l(\tau)$ efficiently represent the remaining information in imaginary-time quantities.

---

[2]As a compact basis for imaginary-time Green's functions, the Legendre basis [22] has been widely used in the context of DMFT and QMC. This fact is interesting because it shows that the Legendre basis corresponds to the high-temperature limit of the IR basis.

[3]Note that the analytical form of the IR basis has yet to be determined as well. If you are good at mathematics, please give it a try.

[4]The convergence of $\rho_l$ requires to establish $\exists V_{\max} \in \mathbb{R} \; \forall l \in \mathbb{N} : ||V_l||_\infty \leq V_{\max}$. This is a conjecture supported by numerical evidence.

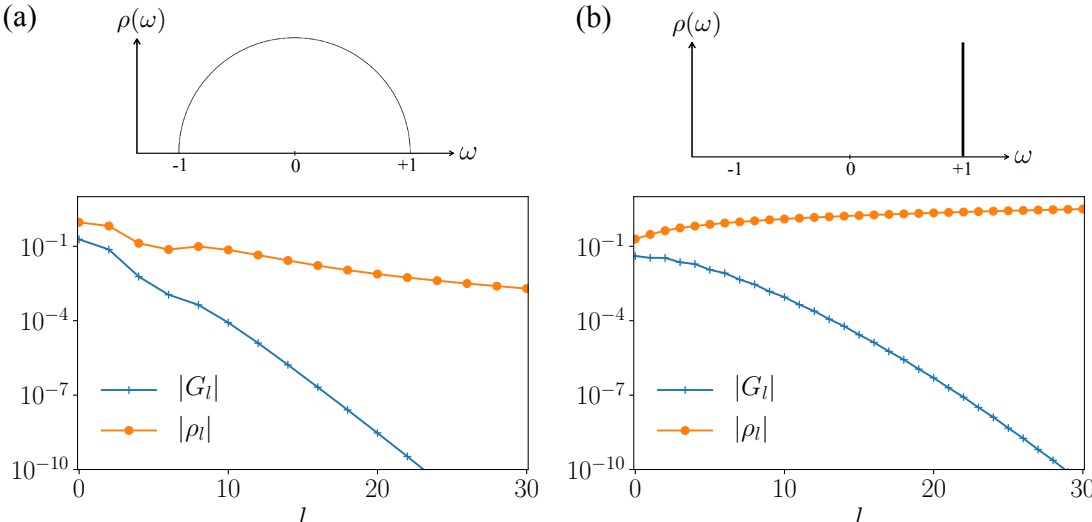

Figure 2: IR expansion coefficients $G_l$ (Green's function) and $\rho_l$ (spectral function) for (a) a semicircular spectral function and (b) a delta function. We take $\beta = 100$, $\omega_{\max} = 1$, and $\alpha = \mathrm{F}$. In (a), $G_l$ and $\rho_l$ for an odd $l$ are not plotted because they are values of zero.

Finally, we present a numerical example that demonstrates the advantages of the IR basis at low temperatures. Figure 3 shows the $\beta$-dependence of the number of expansion coefficients, $N$, required to reconstruct the Green's function within a given accuracy for the model in Fig. 2(b). Three kinds of expansions are compared: the IR-basis expansion, the Legendre expansion, and the Matsubara (imaginary-frequency) expansion. As expected from Fig. 1(a), $N$ increases logarithmically in the IR basis. By contrast, in the cases of the imaginary-frequency representation and the Legendre basis [22], $N$ scales as $O(\beta)$ and $O(\sqrt{\beta})$, respectively. The advantage of the IR basis becomes more pronounced at lower temperatures.

## 3 Sparse sampling

### 3.1 Dyson equation

In the previous section, we showed that the Matsubara Green's function can be represented compactly using the IR basis. A natural question is whether we can efficiently solve diagrammatic equations in the imaginary-time formalism. As an example, let us consider the Dyson equation defined as

$$\hat{G}(i\omega) = \frac{1}{i\omega + \mu - \mathcal{H} - \hat{\Sigma}(i\omega)} \,. \tag{11}$$

The self-energy $\hat{\Sigma}(i\omega)$ can also be compactly expanded with the IR basis:

$$\hat{\Sigma}(i\omega) \simeq \sum_{l=0}^{L-1} \Sigma_l \hat{U}_l^{\mathrm{F}}(i\omega). \tag{12}$$

Here, we excluded the Hartree term for simplicity. The question is whether the expansion coefficients of Eq. (11), $G_l$, can be computed efficiently from the given $\Sigma_l$. In the following demonstration, for simplicity, we set $\mathcal{H} = 0$ and $\mu = 0$.

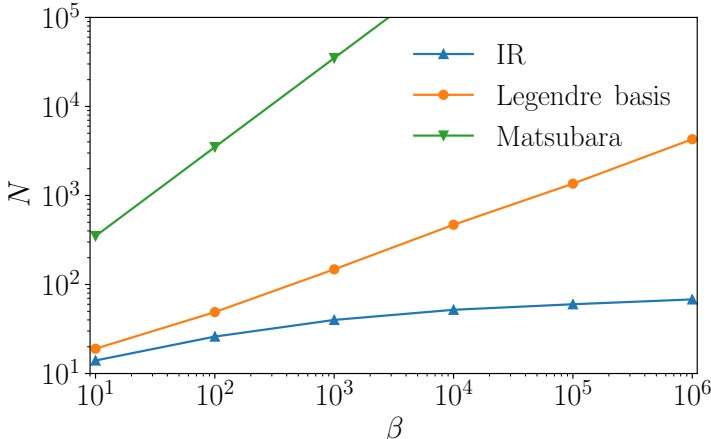

Figure 3: $\beta$–dependence of the number of basis functions, $N$, needed to reconstruct the Green's function with a high accuracy. We used the same spectral function as in Fig. 2(b). We counted the minimum number of basis functions required to represent $G(\tau = 0)$ with an accuracy of $10^{-8}$. For the the imaginary-frequency representation, we count the number of nonnegative imaginary frequencies. A second-order high-frequency expansion is applied in the Fourier transform into $G(\tau)$.

Solving the Dyson equation directly in the IR basis is not the most computationally efficient method. To see this, we transform the Dyson equation (11) into a linear equation for $\hat{G}(i\omega)$ as

$$\sum_{i\omega'} A_{i\omega,i\omega'} \hat{G}(i\omega') = 1 \, . \tag{13}$$

Here, the linear operator $A$ is represented as a diagonal matrix defined by $A_{i\omega,i\omega'} \equiv \delta_{i\omega,i\omega'}(i\omega - \hat{\Sigma}(i\omega))$. In principle, we can transform $A_{i\omega,i\omega'}$ and $\hat{G}(i\omega')$ into the IR basis. However, in the transformed linear equation, the linear operator is no longer diagonal, and we need to solve the linear equation of size $L \times L$. Therefore, the computational cost of the Dyson equation scales as $O(N_{\mathrm{band}}^3 L^3) = O(N_{\mathrm{band}}^3 (\log(\beta\omega_{\mathrm{max}}))^3)$ for multiband systems. This computational complexity grows more slowly than any power of $\beta\omega_{\mathrm{max}}$. Although this is asymptotically faster than the conventional approach, this method is not advantageous for typical parameters in *ab initio* calculations, $N_{\mathrm{band}} = 100$ and $L \simeq 100$.

How can we simultaneously benefit from the compactness of the representation and the diagonality of the Dyson equation? We will answer this question in the next section.

## 3.2 Sparse sampling and Fourier transform

The sparsity of information can be exploited to reduce the computational complexity of the diagrammatic equations either by the sparse sampling method [12] or the minimax isometry method [24]. Before introducing the sparse sampling method, we discuss how to extract information efficiently from physical quantities on the imaginary frequency axis.

The inverse transformation of Eq. (9) is given as

$$G_l = \sum_{i\omega=-i\infty}^{i\infty} \hat{U}_l^{\mathrm{F}}(i\omega)^* \hat{G}(i\omega). \tag{14}$$

Here, we ignore $\epsilon_L$ in Eq. (9). In principle, to determine $G_l$ from this equation, we need to know $\hat{G}(i\omega)$ in the whole imaginary-frequency domain. However, we do not need to evaluate

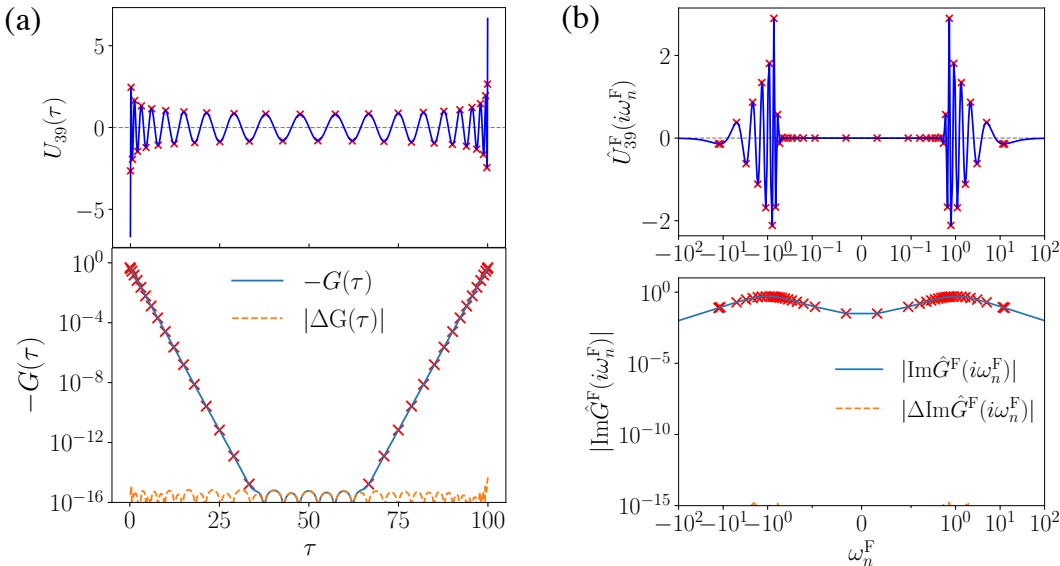

Figure 4: (Upper panel) The basis function at $l = 39$ and the sampling points (red crosses) determined from the basis function. (Lower panel) The Green's function reconstructed by sparse sampling (solid line) and the deviation from the exact value (dashed line). The exact Green's function is not shown because it is almost identical to the reconstructed value. (a) Imaginary time domain and (b) imaginary frequency domain. We used the semicircular model shown in Fig. 2(a) ($\beta = 100$, $\omega_{\max} = 1$). Revised version of Fig. 2 in Ref. [12].

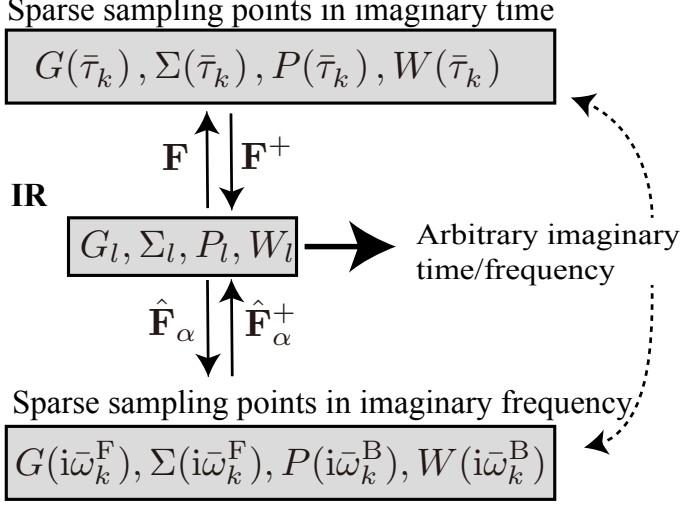

Figure 5: Overview of the sparse sampling method. One can efficiently transform data between sparse meshes in the imaginary frequency and time domains through the IR basis. The data can be evaluated at any frequency and time once the IR expansion coefficients are known.

this equation in practice because the frequency dependence of $\hat{G}(i\omega)$ has only $L$ degrees of freedom at most, as shown in Fig. 2. Therefore, if we choose the appropriate $L$ imaginary frequencies $i\omega$ and know $\hat{G}(i\omega)$ at those points, we can determine the values of $G_l$ for all $l = 0, \cdots, L-1$. We call these frequencies "sampling frequencies."

The selection of sampling frequencies is not unique. In particular, we use the fact that $\hat{U}_l^\alpha(i\omega)$ has a sign structure similar to a polynomial, and choose the maximum and minimum neighborhoods of $\hat{U}_{L-1}^\alpha(i\omega)$ (see Fig. 4):

$$\mathcal{W}^\alpha = \{i\bar{\omega}_1^\alpha, i\bar{\omega}_2^\alpha, \ldots, i\bar{\omega}_L^\alpha\} \,. \tag{15}$$

Given the values of the Green's function on the sampling frequencies, we can easily calculate $G_l^\alpha$ using the least-squares fitting [5] as

$$\begin{aligned}
G_l &= \underset{G_l}{\text{argmin}} \sum_{i\omega \in \mathcal{W}^\alpha} \left| \hat{G}(i\omega) - \sum_{l=0}^{L-1} \hat{U}_l^\alpha(i\omega) G_l \right|^2 \\
&= \left( \hat{\mathbf{F}}_\alpha^+ \hat{\boldsymbol{g}} \right)_l \,.
\end{aligned} \tag{16}$$

Here, $\hat{\mathbf{F}}_\alpha^+$ is the Moore-Penrose pseudo-inverse of $\hat{\mathbf{F}}_\alpha$, where $\hat{\mathbf{F}}_\alpha$ is a matrix consisting of the values of the basis functions at the sampling frequencies: $(\hat{\mathbf{F}}_\alpha)_{kl} = \hat{U}_l^\alpha(i\bar{\omega}_k^\alpha)$. By contrast, $\hat{\boldsymbol{g}}$ is a vector consisting of the values of the Green's function to be fitted: $(\hat{\boldsymbol{g}})_k = \hat{G}^\alpha(i\bar{\omega}_k^\alpha)$. Because $\hat{\mathbf{F}}_\alpha$ is a small matrix of at most $100 \times 100$, as long as the sampling frequencies are chosen as described above, the fitting procedure is numerically stable (refer to Sec. 3.3 and Appendix B). Once $\hat{\mathbf{F}}_\alpha^+$ is computed, Eq. (16) is a matrix-vector product: The computational complexity is $O(L^2)$, and fast libraries such as BLAS can be used.

Similar sampling points in imaginary time can also be constructed by considering the sign structure of $U_l(\tau)$. For the convention in Eq. (4), one can use the same sampling points for fermions and bosons:

$$\begin{aligned}
G_l &= \underset{G_l}{\text{argmin}} \sum_k \left| \hat{G}(\bar{\tau}_k) - \sum_{l=0}^{L-1} \hat{U}_l(\bar{\tau}_k) G_l \right|^2 \\
&= \left( \mathbf{F}^+ \boldsymbol{g} \right)_l \,,
\end{aligned} \tag{17}$$

where $\bar{\tau}_k$ are the sampling points and $(\mathbf{F})_{kl} \equiv U_l(\bar{\tau}_k)$.

Now, let us numerically demonstrate the accuracy of the sparse sampling method. The upper panels in Figs. 4(a) and (b) show the IR basis functions and the sampling points at $\omega_{\text{max}} = 1$ and $\beta = 100$. It can be clearly seen that the basis functions have sign changes. The imaginary-time sampling points are densely distributed near $\tau = 0$ and $\beta$, where $G(\tau)$ varies significantly [see the upper panel in Fig. 4(a)]. In the imaginary-frequency domain, the distribution of sampling points is sparse at high frequencies where the Green's function essentially has no structure [see the upper panel in Fig. 4(b)]. The lower panels in Figs. 4(a) and (b) show the Green's functions reconstructed from $G_l$. The reconstructed Green's functions agree with the exact values with an accuracy of approximately 15 digits, indicating the accuracy and numerical stability of the sparse sampling method.

Some may wonder why we can determine the expansion coefficient $G_l$ of $l < L-1$ accurately using the sampling points determined from the structure of $U_{L-1}(\tau)$ and $U_{L-1}^\alpha(i\omega)$. In fact, the distribution of the roots (zeros) of the IR basis functions for $l < L-1$ is always sparser than that of the basis functions for $l = L-1$ (Figure 1)[6]. Therefore, the sampling

---

[5]Note that the $L_2$ norm is not only the choice in the fitting. It is claimed that using the infinite norm leads to smaller errors [25].

[6]More precisely, for $l' < l$, between two adjacent roots of $U_{l'}(\tau)$, $U_l(\tau)$ contains one or more roots. In the case of $l = l' + 1$, this is mathematically proven. In a general case, no counterexample was found in the numerical experiments.

Table 1: Comparison between the sparse sampling method and the conventional method (FFT). The power $\gamma$ ($\geq 1$) depends on the details of the high-frequency expansion.

|  | Memory cost | Truncation error | Runtime cost |
|---|---|---|---|
| Sparse sampling | $L = O(\log(\omega_{\max}\beta))$ | $O(e^{-\alpha L})$ | $O(L^2)$ |
| Conventional method (FFT) | $N_\omega = O(\omega_{\max}\beta)$ | $O(1/N_\omega^\gamma)$ | $O(N_\omega \log N_\omega)$ |

points determined from $U_{L-1}(\tau)$ and $U_{L-1}^\alpha(i\omega)$ capture the sign changes of the basis functions for $l < L-1$. For this reason, they serve as good sampling points for the truncated basis of size $L$.

We now have all ingredients of the sparse sampling method. Henceforth, the sampling points at imaginary frequency and imaginary time will be referred to as the "sparse mesh"[7]. The sparse sampling method is summarized in Fig. 5. We can efficiently transform the data on the sparse mesh (either in imaginary time or imaginary frequency) to the IR basis, and vice versa. Once the expansion coefficients $G_l$ in the IR basis are obtained, the Green's function can be reconstructed at any frequency or time. In this sense, sparse sampling is a physically motivated interpolation method for imaginary-time and imaginary-frequency quantities.

The sparse sampling method allows us to solve the Dyson Eq. (21) by considering only the sampling frequencies. The exact imaginary frequency/time dependence can be reproduced from these values on the sparse mesh.

Traditionally, when one wants to transform data between imaginary frequency and time, the fast Fourier transform is the standard method. Table 1 shows a comparison between the conventional method and the sparse sampling method. As the main feature of the sparse sampling method and based on the data size, the computational complexity increases more slowly than any power of $\beta$ and bandwidth $\omega_{\max}$. Note that the sparse sampling method can also be used in combination with conventional polynomial bases [12].

### 3.3  Notes on calculations using the sparse sampling method

In practical applications using the sparse sampling method, one should make sure if $\omega_{\max}$ and $L$ are large enough. If $\omega_{\max}$ or the basis size $L$ is not large enough, this may introduce large systematic errors in the basis representation as defined in Eq. (9), such as large truncation errors $\epsilon_L$. Such a systematic error may be amplified in the "fitting" procedure of Eqs. (16) and (17), leading to a large numerical error. Therefore, to ensure stable numerical calculations with the sparse sampling method, one should choose the appropriate basis parameters $L$ and $\Lambda$ such that the basis representation stays accurate. The choice of the basis parameters ($L$, $\Lambda$) can be verified by checking if expansion coefficients $G_l$ decay as fast as the singular values down to the noise level. Otherwise, one should increase the value of $\Lambda$, e.g., by a factor of 10 or increase $L$. The basis size $L$ must be chosen so that the $S_{L-1}/S_0$ is at least the desired accuracy for results multiplied by the condition number of the fitting matrix.

### 3.4  Related approaches

After establishing the idea of compressing the finite-$T$ Green's function using the spectral representation in [10], related approaches have been proposed. In this susbsection, we give a brief overview on these approaches.

M. Kaltak and G. Kresse independently proposed a similar sampling method [24] by extending previously proposed zero-$T$ optimal time and frequency grids [26, 27] to finite tem-

---

[7]The minimax isometry method [24] uses a similar sparse mesh.

perature. Their choice of sampling times and frequencies is based on a different principle (minimax isometry method). One interesting difference from the sparse sampling is that their sampling points are not associated with an orthogonal basis.

An interesting follow-up study of IR has been done recently by J. Kaye *et al.* [20]. They proposed to use simple exponentials in the time domain corresponding to $\delta$ functions in the real-frequency domain as well as a systematic approach to choose sampling frequencies and times. Their "discrete Lehmann representation (DLR)" is generated by essentially the same idea as of the IR, i.e., on a low rank decomposition of the kernel. DLR is slightly less compact than IR and non-orthogonal, but has the same scaling with respect to $\beta \omega_{\mathrm{max}}$. IR is unique for a given kernel and $\beta$, while DLR depends on the pivoting scheme and is not unique.

A detailed comparison of these related approaches would be an interesting topic for future study.

## 3.5 Discrete Lehmann representation (DLR)

We explain the implementation of the DLR in our library. The poles on the real-frequency axis selected for the DLR are based on a rank-revealing decomposition, which offers accuracy guarantees. In our library, we instead select the pole locations based on the zeros of the IR basis functions on the real axis. This is a heuristic motivated by an observation in a compact discretization of an electron bath of a quantum impurity model [28]. We do not expect that difference to matter, which is however to be confirmed in a future study.

We first model the spectral function as

$$\rho(\omega) = \sum_{p=1}^{L} c_p \delta(\omega - \bar{\omega}_p), \tag{18}$$

where sampling frequencies $\{\bar{\omega}_1, \cdots, \bar{\omega}_L\}$ are chosen to be the extrema of $V'_{L-1}(\omega)$. This choice is heuristic but allows us a numerically stable transform between $\rho_l$ and $c_p$ through the relation

$$\rho_l = \sum_{p=1}^{L} V_{lp} c_p, \tag{19}$$

where the matrix $V_{lp}$ $[\equiv V_l(\bar{\omega}_p)]$ is well-conditioned. Figure 6(a) shows the sampling frequencies generated for $\beta = 100$ and $\omega_{\mathrm{max}} = 1$ ($L = 40$).

It is clear that for any given $\{g_l\}$ ($l = 0, \cdots, L-1$), one can find $\{\rho_l\}$ ($l = 0, \cdots, L-1$) that approximates the Green's function within a desired accuracy by minimizing $\sum_{l=0}^{L-1} |g_l + S_l \rho_l|^2$. Then, the pole coefficients $\{c_p\}$ are obtained from $\{\rho_l\}$ using Eq. (19).

Figures 6(b) and 6(c) demonstrate the accuracy of DLR for the semicircular-DOS model [see Fig. 2(a)]. Figure 6(b) shows the expansion coefficients in DLR, $c_p$, computed by fitting the IR expansion coefficients $G_l$. The DLR expansion coefficients $c_p$ exhibits a dip around $\omega = 0$ where the distribution of the sampling frequencies is dense. As shown in Fig. 6(c), the IR coefficients reconstructed from the DLR coefficients match the numerically exact values precisely.

Fitting a Green's function with such a pole basis is equivalent to the infamously difficult analytic continuation problem, i.e., reconstructing a spectral function. If the input is noisy, one needs an appropriate regularization.

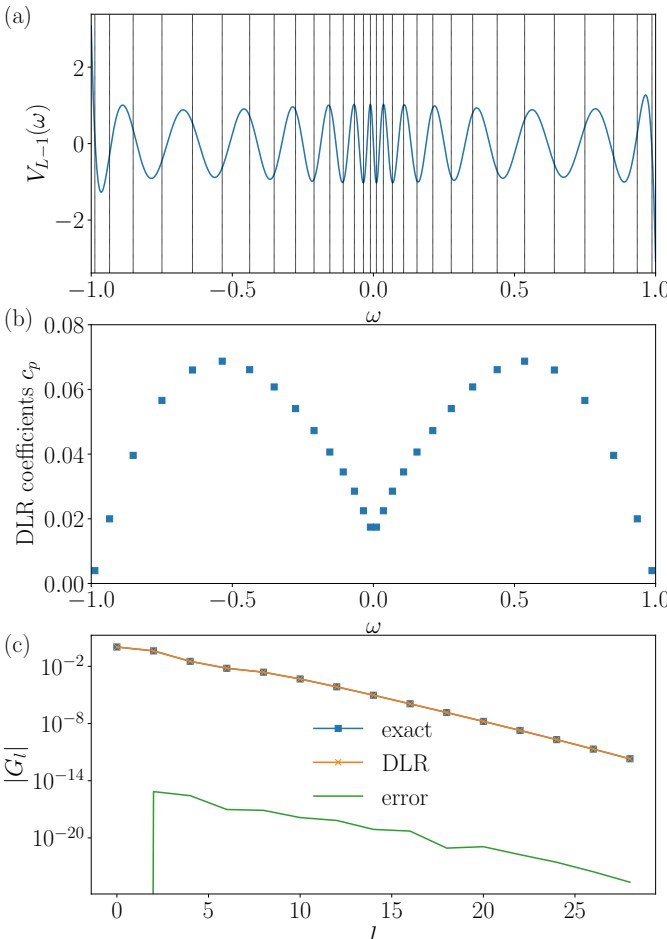

Figure 6: Discrete Lehmann representation (DLR). (a) $V_{L-1}(\omega)$ computed for $\beta = 100$ and $\omega_{\max} = 1$ (fermion, $L = 40$). The vertical lines denote sampling frequencies. (b) Expansion coefficients in DLR for the Green's function of the semi-circular DOS model [see Fig. 2(a)]. (c) Comparison of IR expansion coefficients reconstructed from the DLR data in (b) and the numerically exact result.

## 4 Review of applications

In the previous section, we showed that the expansion coefficients of the IR basis functions can be calculated with high accuracy from the values of Green's functions for a small number of sampling points. This makes it possible to efficiently transform data between the imaginary-time and imaginary-frequency domains. In this section, we show how the sparse sampling method can be used to efficiently solve the diagrammatic equations.

### 4.1 Noble-gas atoms, silicon crystals: Self-consistent $GW$ and second-order perturbation (GF2) calculations

From Ref. [12], which proposed a sparse sampling method, we present the results of a self-consistent $GW$ calculation and a benchmark calculation using second-order perturbations (GF2). The purpose of this study was to verify the stability and accuracy of the numerical calculations. Thus, they chose the physically "trivial" but technically difficult noble gas atoms and silicon crystals. In this article, we present the results for noble gases. Please refer to the original paper [12] for the results for silicon crystals.

We consider the Hamiltonian

$$H = \sum_{ij\sigma} h_{ij} c_{i\sigma}^\dagger c_{j\sigma} + \frac{1}{2} \sum_{ijkl} \sum_{\sigma\sigma'} V_{ijkl} c_{i\sigma}^\dagger c_{k\sigma'}^\dagger c_{l\sigma'} c_{j\sigma} , \tag{20}$$

where $h_{ij}$ is the non-interacting Hamiltonian; $V_{ijkl}$ is the Coulomb integral; $i$, $j$, $k$, and $l$ are orbital indices; and $\sigma$ and $\sigma'$ are spin indices. In this case, the Dyson equation reads:

$$\hat{G}(i\omega_n^{\mathrm{F}}) = [(i\omega_n^{\mathrm{F}} + \mu)I - F - \hat{\tilde{\Sigma}}(i\omega_n^{\mathrm{F}})]^{-1} , \tag{21}$$

where the Fock matrix is given by $F = h + \Sigma^{\mathrm{HF}}$. The Hartree-Fock term can be computed as

$$\Sigma_{ij}^{\mathrm{HF}} = (2V_{ijkl} - V_{ilkj})\rho_{kl} , \tag{22}$$

where we used Einstein's notation. We use different $\hat{\tilde{\Sigma}}(i\omega_n^{\mathrm{F}})$ for different approximations. Based on the GF2 theory, we use

$$\begin{aligned}
\tilde{\Sigma}_{ij}(\tau) \simeq \Sigma_{ij}^{(2)}(\tau) &= -G_{kl}(\tau)G_{qm}(\tau)G_{np}(-\tau) \\
&\times V_{ikpq}(2V_{ljmn} - V_{mjln}) .
\end{aligned} \tag{23}$$

In the *GW* approximation, we take

$$\tilde{\Sigma}_{ij}(\tau) \simeq \Sigma_{ij}^{GW}(\tau) = -G_{lk}(\tau)\tilde{W}_{ilkj}(\tau) . \tag{24}$$

Here, $\tilde{W} = W - V$ is the dynamic part of the screened interaction $W$:

$$\hat{W}_{ijkl}(i\omega_n^{\mathrm{B}}) = V_{ijkl} + V_{ijpq}\hat{P}_{qpsr}(i\omega_n^{\mathrm{B}})\hat{W}_{rskl}(i\omega_n^{\mathrm{B}}) . \tag{25}$$

The bare polarization function is given by

$$P_{ijkl}(\tau) = -G_{il}(\tau)G_{jk}(-\tau) . \tag{26}$$

Equations (21)–(26) are diagonal in the imaginary frequency or imaginary time. Therefore, if the IR expansion coefficients of the response functions appearing on the right-hand side of each of are known, the values of the left-hand side can be calculated quickly on the sampling points and then immediately converted into the IR basis. Thus, by going through the IR basis, we can conduct self-consistent calculations while keeping the data compact.

Figure 7 shows the results of the calculation using the all-electron basis (cc-pVDZ) for noble-gas atoms ($\beta = 1000 E_{\mathrm{h}}^{-1}$, $T \simeq 316$ K). Owing to the deep core state of noble gas atoms, it is difficult to apply all-electron calculations on a conventional polynomial basis, which requires thousands of basis functions even with the introduction of an effective potential. However, by combining the IR basis with sparse sampling, we can conduct all-electron calculations with as few as 100 basis functions.

As the number of basis functions $L$ increases, the total energy converges exponentially, with relative errors easily converging below the level of $10^{-10}$. For heavier noble-gas atoms, the core states deepen. The IR basis can easily handle these deep core states by increasing $\Lambda = \beta \omega_{\max}$. The results of the *GW* method, which involves the treatment of the response functions of both fermions and bosons, also show convergence comparable to that of GF2. This indicates the high numerical stability of the method.

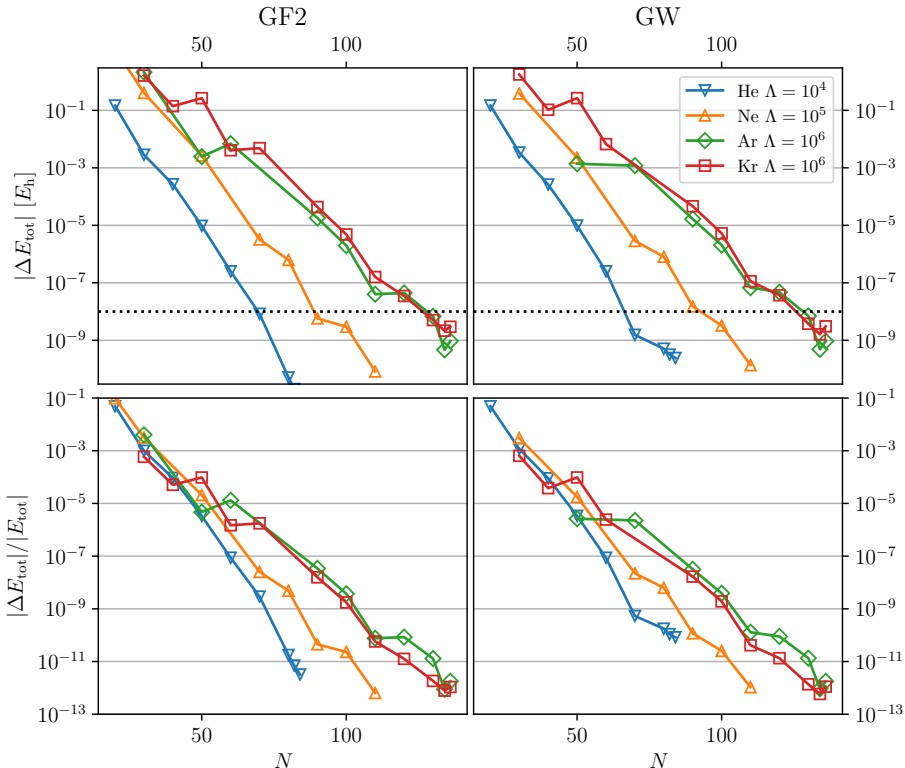

Figure 7: Convergence of total energy in GF2 (left panel) and *GW* (right panel) calculations for noble gas atoms. The horizontal axis is the number of IR bases. The upper panel shows the absolute error of the energy (in Hartree $E_h$), and the lower panel shows the relative error. From Ref. [12].

## 4.2 First-principles calculations of superconducting transition temperature based on Migdal–Eliashberg equation

The transition temperature ($T_c$) of a conventional (phonon-mediated) superconductor was recently obtained from first-principles calculations. The main theoretical approaches are the superconducting density functional theory and Migdal–Eliashberg theory. The latter has the advantage that the effective mass renormalization—owing to the electron-phonon interaction—can be determined self-consistently. By contrast, a realistic prediction of the transition temperature is extremely difficult at low temperatures because it requires simultaneous treatment of a phonon-mediated attraction and Coulomb repulsion, the energy scales of which are two or three orders of magnitude apart. In this section, we introduce a recent study [29], which tackles this problem using the IR basis.

In the framework of the Migdal-Eliashberg theory, $T_c$ is obtained by solving the following linearized gap equation:

$$\tilde{\lambda}\Delta_i(i\omega_n^{\mathrm{F}}) = -T \sum_{j,\omega_{n'}^{\mathrm{F}}} K_{ij}(i\omega_n^{\mathrm{F}} - i\omega_{n'}^{\mathrm{F}})F_j(i\omega_{n'}^{\mathrm{F}}). \tag{27}$$

Here, subscript $i$ denotes a combined index of the momentum and the energy band, $\Delta_i$ is the superconducting gap function, and $K_{ij} = K_{ij}^{\mathrm{el-ph}} + K_{ij}^{\mathrm{el-el}}$ denotes the electron-phonon and (screened) Coulomb interaction. Because the anomalous Green's function $F_i$ is given by

$$F_i(i\omega_n^{\mathrm{F}}) = |G_i(i\omega_n^{\mathrm{F}})|^2 \Delta_i(i\omega_n^{\mathrm{F}}). \tag{28}$$

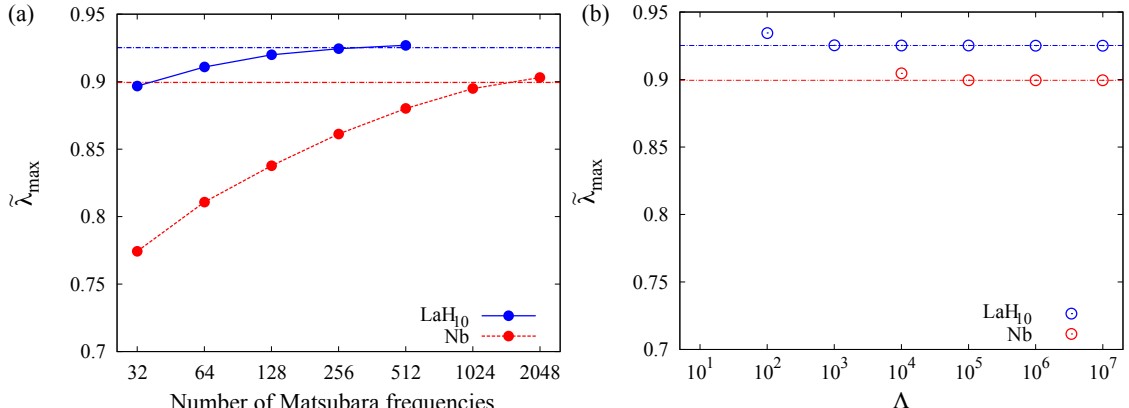

Figure 8: The convergence of the maximum eigenvalue $\tilde{\lambda}_{max}$ of the linearized gap equation is just above the transition temperature. (a) Imaginary frequency + FFT and (b) IR basis + sparse modeling. The chain line in the figure shows the value obtained using the IR basis. This plot is essentially the same as Fig. 2 of Ref. [29]. The minor differences in the values of $\tilde{\lambda}_{max}$ originate from differences in implementation details.

Equation (27) is an eigenvalue problem for a matrix, and $\tilde{\lambda}$ is its eigenvalue. The general framework of the theory aims to solve Eq. (27) using $G_i$, including the self-energy correction by $K_{ij}^{el-ph}$, to find $T_c$ as the temperature at which the maximum value of $\tilde{\lambda}$ reaches 1.

Because the convolution integrals of the boson and fermion Green's functions are diagonal in imaginary time, the IR basis and the sparse sampling method can be applied to this problem. In general, because realistic materials contain a large number of bands, and the number of $k$-points required for convergence is large at low temperatures, it is difficult to solve Eq. (27). Using the IR basis, we can expect a dramatic improvement, as shown in Table 1.

The actual eigenvalues of the linearized gap equation $\tilde{\lambda}$ for BCC-Nb and cubic LaH$_{10}$ under high pressure (250 GPa) are shown in Fig. 8. The calculated temperatures are 19.7 and 271.6 K, respectively, which are slightly higher than the respective transition temperatures. Figure 8(a) shows the conventional method (imaginary frequency + FFT), which converges roughly at $N_\omega = 512$ for LaH$_{10}$, but still does not converge at $N_\omega = 2048$ for Nb. As a rough estimate, if we assume that the upper limit of the band is $\omega_{max} \sim 10$ eV, which leads to $\beta\omega_{max} \simeq 10^4$, we will need a higher $N_\omega$ to converge for Nb. However, with the IR basis + sparse sampling method shown in Figure 8(b), we already have a fully converged value at $\Lambda = 10^4$. Comparing the two calculations with $N_\omega = 4096$ (shown in Ref. [29]) and $\Lambda = 10^5$ as an example, we found that the number of bases is $L = 138$, which saves approximately 30-times the amount of memory and improves the actual calculation speed by approximately 20-fold. We note that we did not successfully obtain converged self-consistent solutions for $\Lambda < 10^2$ (LaH$_{10}$) and $\Lambda < 10^4$ (Nb), which may be because $\Lambda$ was not large enough (refer to Appendix B). In the present calculation, the static shielding Coulomb interaction is used as the electron-electron interaction, but it can be extended to include the plasmon effect in the random phase approximation.

## 4.3 Calculation of magnetic interaction based on Liechtenstein formula

As is well known, density functional theory (DFT) is applicable only to the ground state, and it is impossible to calculate the properties of finite temperatures without modification. In the case of magnetic materials, the magnitude of the magnetic moment at absolute zero can be predicted; however, the transition temperature cannot be calculated. Although there are several approaches to this problem, the most convenient and commonly used approach is to

derive an effective spin model from the results of the electronic structure calculation, and to obtain the transition temperature by a calculation using the spin model (*e.g.*, mean-field approximation and Monte Carlo calculation). Among the mappings to the spin model [30], a method called the Liechtenstein method, has been found to significantly improve the efficiency of the calculation by using the IR basis + sparse sampling method, which is briefly introduced in this subsection [13–15].

In this theory, we compute the interaction parameters $J_{ij}$ in the Heisenberg-type spin model $H = -2 \sum_{\langle i,j \rangle} J_{ij} S_i \cdot S_j$ as follows:

$$J_{ij} = -\frac{T}{2} \sum_{\omega_n} \mathrm{Tr}\Big[ G_{ij}(i\omega_n^{\mathrm{F}}) M_j G_{ji}(i\omega_n^{\mathrm{F}}) M_i \Big]. \tag{29}$$

Here, $G_{ij}$ is Green's function for an effective tight-binding model derived from DFT calculations, and $i, j$ are site indices. When the number of orbitals belonging to site $i$ is $N_{\mathrm{orb}}^i$, $G_{ij}$ is an $N_{\mathrm{orb}}^i \times N_{\mathrm{orb}}^j$ matrix. In addition, although $M_i$ is a perturbation term to the Hamiltonian owing to the infinitesimal rotation of the spins, here it should simply be regarded as a constant matrix of $N_{\mathrm{orb}}^i \times N_{\mathrm{orb}}^i$, which does not depend on $\omega_n$. Moreover, Tr in Eq. (29) denotes the trace regarding the orbital component. Considering the computational cost required to evaluate this term, the dominant factor is the Fourier transform of the Green's function from the $k$ space to the real space, which is approximately $O(cN_\omega)$ using the usual Matsubara frequency mesh. Here, the coefficient factor $c$ is approximately $c = O((N_k \log N_k)^3 N_{\mathrm{orb}}^2)$ when the number of $k$ meshes in one dimension is $N_k$. By contrast, if Green's function calculation is limited to the sparse meshes, the computational cost becomes approximately $O(cL)$, which is much lower at low temperatures. In the actual calculation, the trace of Eq. (29) is calculated on the sparse mesh and finally converted into the IR basis. Subsequently, Eq. (29) is obtained by recovering the values at the imaginary time $\tau = 0$ from the expansion coefficients in the IR basis. In this method, the loss in the basis conversion is almost negligible, and the computation time can be reduced by a factor of $N_\omega / L$.

In this study, although we used the Liechtenstein method as an example, we can generally apply this method for the free energy $F[A]$ of a system described by the second-order Hamiltonian $H = \sum_{12} A_{12} c_1^\dagger c_2$, with the derivative $\delta_{A_{12}} F$, $\delta_{A_{12}} \delta_{A_{34}} F$, $\cdots$ with respect to $A$. Although this computation is not extremely heavy, it is a good example showing that the IR basis can be used as a common tool. As we have seen, the convergence of the calculations using the IR basis is extremely high, and the number of parameters required to check the convergence can be reduced, which is always a significant advantage in research using numerical calculations.

## 4.4 Other applications

In addition to those introduced in this section, there are many other applications in the field of *ab initio* calculations, such as the calculation of the superconducting transition temperature using the FLEX approximation [17,18], *ab initio* calculations of NiO compounds based on the self-energy embedding method [16], and an *ab initio* estimation of the magnetic interaction constant of $NdNiO_2$ [15]. The exponential convergence of the calculation accuracy with respect to the basis size is also useful in model calculations, such as self-energy denoising [31] and efficient discretization of quantum impurity problems [28].

## 5 `sparse-ir` library

In the previous sections, we described the IR basis and sparse sampling method. We have developed the Python library `sparse-ir` [32], which is an updated version of `irbasis` [33]. This

library can be used from Julia and Fortran as well. This library allows these new techniques to be easily used without knowing the details of the computation of the IR basis functions and sampling points. Once the values of the basis functions and the sampling points are written to a file, diagram calculations using sparse sampling can be conducted in any programming language (the superconductivity calculations described above were actually implemented in Fortran).

sparse-ir [32], has a more user-friendly interface than its predecessor irbasis [33]. While irbasis stores (precomputed) tabulated data of the IR basis functions for several values of $\Lambda$, sparse-ir allows on-the-fly computation of the IR basis functions at an arbitrary value of $\Lambda$. The technical details will be explained in more depth in a follow-up paper.

## 5.1 Installation

Because sparse-ir is registered to PyPI, its installation is quite simple and can be achieved by running the following command from a shell:

```
$ pip install -U sparse-ir xprec
```

The -U option aims to update to the latest version of sparse-ir if it is already installed. Though this is not strictly required, we strongly recommend installing the xprec package alongside sparse-ir as it allows to compute the IR basis functions with greater accuracy.

## 5.2 Usage

sparse-ir can construct a basis object for given $\beta$ and $\omega_{max}$. Below is an example code that construct a basis for $\Lambda = 1000$ (fermion) and $\beta = 100$ then evaluates the basis functions at certain values of $\tau$, $\omega$, and imaginary frequency $n$.

```python
# Compute IR basis for fermions and \Lambda = \beta * \omega_max =
                                      1000
import sparse_ir
import numpy as np

lambda_   = 1000
eps = 1e-8 # cut-off value for singular values
b_xy = sparse_ir.DimensionlessBasis('F', lambda_, eps)

# All singular values
print("singular values: ", b_xy.s)
print("u_0(0.1)", b_xy.u[0](0.1))
print("v_0(0.1)", b_xy.v[0](0.1))

print("n-th derivative of u_l(x) and v_l(y)")
for n in range(1,3):
    u_n = b_xy.u.deriv(n)
    v_n = b_xy.v.deriv(n)
    print(" n= ", n, u_n[0](0.1))
    print(" n= ", n, v_n[0](0.1))

# Compute u_{ln} as a matrix for the first
# 10 non-nagative fermionic Matsubara frequencies
# Fermionic/bosonic frequencies are denoted by odd/even integers.
hatF_t = b_xy.uhat(2*np.arange(10)+1)
print(hatF_t.shape)
```

When running this, you will instantly obtain the following results.

```
singular values:  [6.93676632e-02 6.39029304e-02 4.73525981e-02
   3.78765556e-02
```

```
   2.69709397e-02 1.97919952e-02 1.38987851e-02 9.79170114e-03
   6.77586537e-03 4.66563008e-03 3.18095991e-03 2.15479043e-03
   1.44908097e-03 9.68433898e-04 6.43222579e-04 4.24765651e-04
   2.78944576e-04 1.82210441e-04 1.18411677e-04 7.65700139e-05
   4.92756267e-05 3.15628325e-05 2.01254341e-05 1.27758728e-05
   8.07529293e-06 5.08266091e-06 3.18587477e-06 1.98887606e-06
   1.23669448e-06 7.65992957e-07 4.72633737e-07 2.90529576e-07
   1.77929561e-07 1.08573268e-07 6.60143404e-08 3.99959952e-08
   2.41478607e-08 1.45293028e-08 8.71232105e-09 5.20671801e-09
   3.10136291e-09 1.84126342e-09 1.08960675e-09 6.42730925e-10]
u_0(0.1) 0.2740189634895232
v_0(0.1) 0.7859154340971233
n-th derivative of u_l(x) and v_l(y)
 n=  1 0.04705147674680676
 n=  1 -5.044715458614174
 n=  2 0.48534261358041686
 n=  2 76.97205691155098
(44, 10)
```

One can evaluate the basis functions for multiple values of $x$, $y$, or $n$ with a single function call (corresponding to numpy broadcasting, see the examples below).

sparse-ir can construct a basis object for given $\beta$ and $\omega_{\max}$ directly, allows us to evaluate $U_l(\tau)$, $V_l(\omega)$, $\hat{U}^\alpha(i\omega_n)$ directly. This is more convenient in practical calculations. The interface is quite similar to that for the dimensionless presentation. An example code is shown below.

```
# Compute IR basis for fermions and \beta = 100 and \omega_max = 10
lambda_ = 1000
beta = 100
wmax = lambda_/beta
eps = 1e-8 # cut-off value for singular values
b = sparse_ir.FiniteTempBasis('F', beta, wmax, eps=eps)

x = y = 0.1
tau = 0.5 * beta * (x+1)
omega = wmax * y

# All singular values
print("singular values: ", b.s)
print("U_0(0.1)", b.u[0](tau))
print("V_0(0.1)", b.v[0](omega))

print("n-th derivative of U_l(tau) and V_l(omega)")
for n in range(1,3):
    u_n = b.u.deriv(n)
    v_n = b.v.deriv(n)
    print(" n= ", n, u_n[0](tau))
    print(" n= ", n, v_n[0](omega))

# Compute u_{ln} as a matrix for the first
# 10 non-nagative fermionic Matsubara frequencies
# Fermionic/bosonic frequencies are denoted by odd/even integers.
hatF_t = b.uhat(2*np.arange(10)+1)
print(hatF_t.shape)
```

When running this, you will instantly obtain the following results.

```
singular values:  [1.55110810e+00 1.42891296e+00 1.05883628e+00
   8.46945531e-01
 6.03088545e-01 4.42562468e-01 3.10786283e-01 2.18949094e-01
 1.51512956e-01 1.04326660e-01 7.11284259e-02 4.81825788e-02
 3.24024355e-02 2.16548403e-02 1.43828941e-02 9.49804870e-03
 6.23739033e-03 4.07434932e-03 2.64776559e-03 1.71215756e-03
```

```
 1.10183651e-03 7.05766389e-04 4.50018387e-04 2.85677201e-04
 1.80569039e-04 1.13651753e-04 7.12383254e-05 4.44726207e-05
 2.76533293e-05 1.71281232e-05 1.05684116e-05 6.49643881e-06
 3.97862594e-06 2.42777207e-06 1.47612553e-06 8.94337640e-07
 5.39962581e-07 3.24885087e-07 1.94813421e-07 1.16425754e-07
 6.93485829e-08 4.11719016e-08 2.43643475e-08 1.43719004e-08]
U_0(0.1) 0.038752133451430165
V_0(0.1) 0.2485282820026873
n-th derivative of U_l(x) and V_l(y)
 n=  1 0.00013308167309003305
 n=  1 -0.15952790996681684
 n=  2 2.745512426092119e-05
 n=  2 0.24340701602860684
(44, 10)
```

## 5.3 Example: Second-order perturbation theory

As a simple example, we calculate the self-energy of the two-dimensional Hubbard model within the range of second-order perturbations. The following code is available as a Jupyter notebook [35].

First, for $\beta = 10^3$ and $\omega_{\max} = 10^2$ ($\Lambda = 10^5$), we construct a basis object (with 138 basis functions). The generation of sampling points and the computation of $\hat{\mathbf{F}}_\alpha$ and $\mathbf{F}$ as well as their pseudo-inverse matrices are done by `MatsubaraSampling` and `TauSampling`.

```python
import sparse_ir
import numpy as np
from numpy.fft import fftn, ifftn

beta = 1e+3
lambda_ = 1e+5

wmax = lambda_/beta
eps = 1e-15
print("wmax", wmax)

b = sparse_ir.FiniteTempBasis('F', beta , wmax, eps=eps)
print("Number of basis functions", b.size)

# Sparse sampling in tau
smpl_tau = sparse_ir.TauSampling(b)

# Sparse sampling in Matsubara frequencies
smpl_matsu = sparse_ir.MatsubaraSampling(b)
```

We now compute the Green's function for a band dispersion:

$$\epsilon(\boldsymbol{k}) = -2(\cos k_1 + \cos k_2). \tag{30}$$

Note that the Green' function

$$G(\boldsymbol{k}, i\omega) = \frac{1}{i\omega - \epsilon(\boldsymbol{k})} \tag{31}$$

is computed only on the sampling frequencies.

```python
# Parameters
nk_lin = 64
U, kps  = 2.0, np.array([nk_lin, nk_lin])
nw = smpl_matsu.sampling_points.size
ntau = smpl_tau.sampling_points.size
```

```
# Generate k mesh and non-interacting band energies
nk = np.prod(kps)
kgrid = [2*np.pi*np.arange(kp)/kp for kp in kps]
k1, k2 = np.meshgrid(*kgrid, indexing='ij')
ek = -2*(np.cos(k1) + np.cos(k2))
iw = 1j*np.pi*smpl_matsu.sampling_points/beta

# G(iw, k): (nw, nk)
gkf = 1.0 / (iw[:,None] - ek.ravel()[None,:])
```

At this point, gkf is a two-dimensional array of the number of sampling frequencies × the number of $k$-points. In imaginary time and real spaces, the second-order self-energy is given by

$$\Sigma(\tau, r) = U^2 G^2(\tau, r) G(\beta - \tau, r), \tag{32}$$

where $r$ represents a position in real space. Thus, one can evaluate the self-energy by transforming Green's function from the sampling frequencies to the IR basis and then to the sampling $\tau$ points as follows:

```
# G(l, k): (L, nk)
gkl = smpl_matsu.fit(gkf)

# G(tau, k): (ntau, nk)
gkt = smpl_tau.evaluate(gkl)

# G(tau, r): (ntau, nk)
grt = np.fft.fftn(gkt.reshape(ntau, *kps), axes=(1,2)).\
        reshape(ntau, nk)

# Sigma(tau, r): (ntau, nk)
srt = U*U*grt*grt*grt[::-1,:]

# Sigma(l, r): (L, nk)
srl = smpl_tau.fit(srt)
```

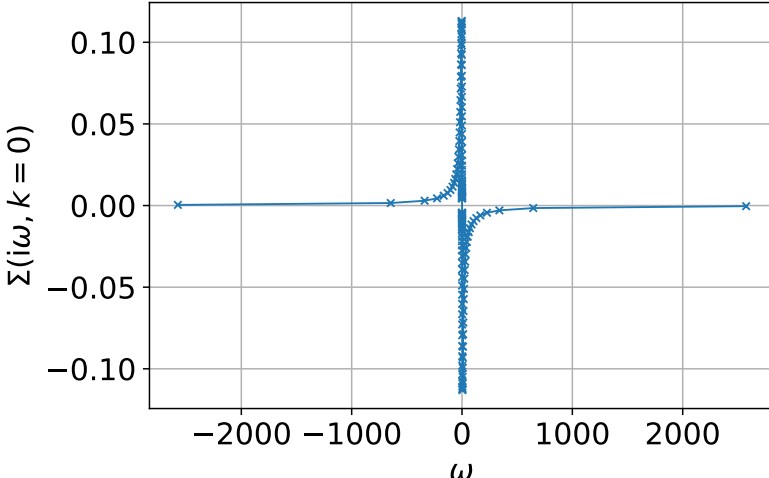

Figure 9: Self-energy computed for the Hubbard model on the sampling frequencies ($k = 0$). In sparse sampling methods, the self-energy is evaluated at a few sampling frequencies.

```
# Sigma(iw, r): (nw, nk)
srf = smpl_matsu.evaluate(srl)
```

At this point, the computed self-energy $\Sigma(\tau, r)$ is stored in a two-dimensional array `srt`. Finally, we transform the self-energy into sampling frequencies (through the IR basis) and momentum space as follows:

```
# Sigma(l, r): (L, nk)
srl = smpl_tau.fit(srt)

# Sigma(iw, r): (nw, nk)
srf = smpl_matsu.evaluate(srl)

# Sigma(iw, k): (nw, kps[0], kps[1])
srf = srf.reshape(nw, *kps)
skf = ifftn(srf, axes=(1,2))/nk**2
```

Figure 9 plots the computed self-energy. The self-consistent calculations can be performed on sampling frequencies/imaginary times without using a dense mesh.

## 6   Summary and future directions

In this Lecture Note, we introduced a compact representation of Green's function and a sparse sampling scheme for solving diagrammatic equations. Using the IR basis, the data size increases only logarithmically with respect to the inverse temperature. Furthermore, the sparse sampling method enables us to efficiently solve the diagrammatic equations, keeping the numerical data compact and numerically accurate. These methods are extremely useful in *ab initio* calculations, where we need to deal with wide bandwidths and low temperatures. As examples, we present the results of *ab initio* calculations using the *GW* approximation and the Migdal-Eliashberg theory. Furthermore, we reviewed the numerical library `sparse-ir`. We hope that this Lecture Note will serve as a starting point for applications in various fields.

Finally, we discuss future directions. The frontier of method developments has already shifted towards two-particle quantities. There are various theories at the two-particle level: diagrammatic calculations with vertex corrections, dynamical susceptibility calculations based on dynamical mean field theory, and nonlocal extensions of dynamical mean field theory that can handle unconventional superconductivity (such as the dual fermion method [36] and the dynamical vertex approximation approach [37]). These two-particle quantities have not only three augments for imaginary frequencies but also arguments for orbitals, spins, and wavenumbers. To carry out calculations using these theories, we need an efficient numerical treatment of two-particle quantities, which remains a challenging issue. The IR basis and the sparse sampling method have been extended to the two-particle quantities to address such problems. The interested readers are referred to recent articles [38–40].

We hope that the development of such fundamental numerical techniques will lead to technological breakthroughs and a better understanding of the properties of strongly correlated compounds based on accurate *ab initio* calculations.

## Acknowledgments

We would like to thank our collaborators in the research described in this article. Our collaboration with Prof. Masayuki Ohzeki, which aims to apply sparse modeling to quantum many-body calculations, led to the discovery of the IR basis.

**Funding information** This work was supported by the following Grant-in-Aid for Scientific Research: No. 15H05885 (J-Physics), No. 16H01064 (J-Physics), No. 18H04301 (J-Physics), No. 18H01158, No. 16K17735, No. 19K14654, No. 19K14654, No. 21H01003, No. 21H01041. HS was supported by JST, PRESTO Grant No. JPMJPR2012. MW acknowledges support by the FWF (Austrian Science Funds) through Project No. P30997. EG and JL were supported by the Simons foundation via the Simons Collaboration on the Many-Electron Problem. TN was supported by JST, PRESTO Grant No. JPMJPR20L7.

# A  Some properties of the IR basis functions

In this appendix, we are justifying some of the properties of the IR basis functions discussed in Sec. 2. Let us do this for the fermionic kernel. For this, we will again write the kernel in terms of dimensionless variables

$$K(x,y) = -\frac{\exp\left(-\frac{\Lambda}{2}xy\right)}{\exp\left(\frac{\Lambda}{2}y\right) + \exp\left(-\frac{\Lambda}{2}y\right)}, \tag{A.1}$$

where again $x = 2\tau/\beta - 1$, $y = \omega/\omega_{\max}$, and $\Lambda = \beta\omega_{\max}$.

This kernel is clearly square integrable, i.e., $\int_{-1}^{1} dx \int_{-1}^{1} dy \, K^2(x,y) < \infty$, since its value is bounded. One can show that any such kernel admits a singular value expansion [41]:

$$K(x,y) = \sum_{l=0}^{\infty} u_l(x) s_l v_l^*(y), \tag{A.2}$$

where again $s_l$ are the singular values and $u_l$ and $v_l$ are the left and right singular functions, respectively.

First, since the kernel is smooth for any temperature, $K \in C^\infty[-1,1]^2$, one has that the singular values decay faster than any power asymptotically [41]. Numerically, one can show that $s_l$ in fact decay faster than exponentially, thus establishing Property 1.

The left and right singular functions in turn are the solution to eigenvalue equations with the following integral kernels:

$$K^u(x,x') = \int_{-1}^{1} dy K(x,y) K(x',y) = \sum_{n=0}^{\infty} \frac{\Lambda^{2n}}{4n+2} \sum_{k=0}^{2n} \frac{E_k(\frac{1}{2}(x+1))E_{2n-k}(\frac{1}{2}(x'+1))}{k!(2n-k)!}, \tag{A.3}$$

$$K^v(y,y') = \int_{-1}^{1} dx K(x,y) K(x,y') = \frac{\tanh\left(\frac{\Lambda}{2}y\right) + \tanh\left(\frac{\Lambda}{2}y'\right)}{\Lambda(y+y')}, \tag{A.4}$$

where $E_n(x)$ is the $n$-th Euler polynomial.

These kernels are real and symmetric by construction, which implies that the singular functions can be chosen to be real. They are also *centrosymmetric*, i.e., one has $K^u(x,x') = K^u(-x,-x')$. Since $s_l$ are not degenerate (see later), this implies that each $u_l(x)$ must be either an odd or an even function in $x$. The same is true for $v_l(y)$. This establishes Property 3.

The kernel $-K(x,-y)$ belongs to the "exponential family" of parametrized probability distributions. These kernels satisfy a property called strict total positivity [42]. We briefly state this property here: The set of $k$-samples $\mathcal{S}_k$ into the space $[-1,1]$ is defined as $\mathcal{S}_k := \{(x_1,\ldots,x_k) : -1 \leq x_1 < \ldots < x_k \leq 1\}$. A kernel $K : [-1,1]^2 \to \mathbb{R}$ is called strictly totally positive (STP) if for any $k \geq 0$ and $(x_1,\ldots,x_k),(y_1,\ldots y_k) \in \mathcal{S}_k$, one has

$$\det \begin{pmatrix} K(x_1,y_1) & K(x_1,y_2) & \ldots & K(x_1,y_k) \\ \vdots & \vdots & \ddots & \vdots \\ K(x_k,y_1) & K(x_k,y_2) & \ldots & K(x_k,y_k) \end{pmatrix} > 0. \tag{A.5}$$

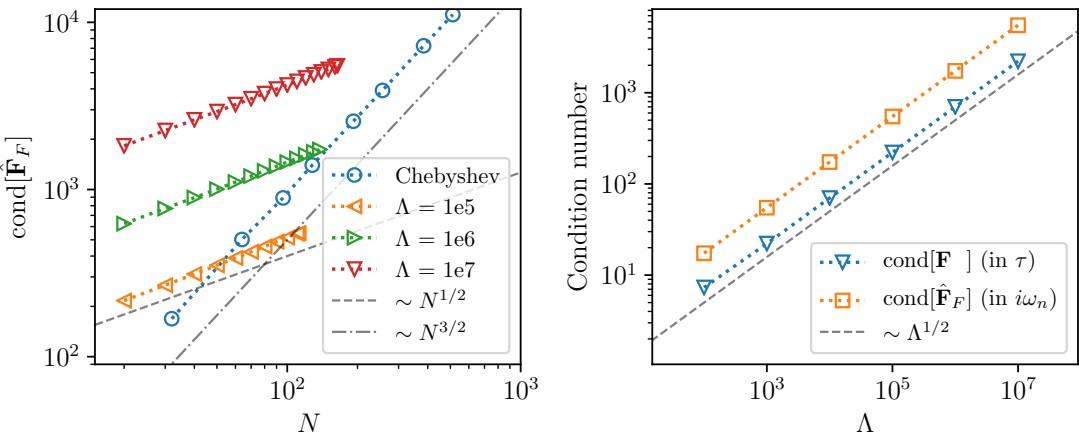

Figure 10: Condition number of the IR transformation matrices $\hat{\mathbf{F}}_F$ and $\mathbf{F}$ [12]. Left panel shows the condition number of frequency transformation matrices $\hat{\mathbf{F}}_F$ as a function of basis size $N = L$, in comparison with the Chebyshev representation. Right panel shows the condition number of both $\tau$ and $i\omega_n$ transformation matrices with respect to $\Lambda$, where $N$ is chosen to be the maximum number of coefficients with the same cutoff in singular values $S_l$, provided in the `irbasis` library [21].

(This generalizes the notion of positive matrices appearing in, e.g., the theory of finite Markov chains, to integral kernels.) One can now show that since $K$ is a STP kernel, then the following holds [42]:

1. there exists a countable set of singular values which are all non-degenerate,

2. the $n$-th left/right singular function has exactly $n$ nodal zeros in $(-1, 1)$ and no zeros of another type,

3. the zeros of the $n$'th and $n+1$'st singular function strictly interlace, i.e., exactly one zero of $u_n$ falls between two consecutive zeros of $u_{n+1}$.

This establishes Property 4.

## B  Numerical stability of sparse sampling

When the expansion coefficients are numerically evaluated using the sparse sampling "fitting" procedures, i.e. Eqs. (16) and (17), numerical errors, such as round-off errors from floating point operations or truncation errors from a finite basis cutoff, may be amplified due to the (pseudo-)inversion process. This error amplification can be quantified by the condition number of the transformation matrices $\mathbf{F}$ and $\hat{\mathbf{F}}_\alpha$, defined as the product of the 2-norms of the matrix and its inverse. In Fig. 10 we show the behaviors of such condition numbers for the IR basis as a function of the basis size $N = L$ (left panel, compared to the Chebyshev representation), and as a function of $\Lambda$ (right panel). We can see that up to a significant number of basis functions, the condition numbers are $< 10^4$, which indicates well-conditioned inversion problems. In addition, the condition numbers show an approximate scaling of $O(L^{1/2})$, which is slower that of the Chebyshev representation $O(L^{3/2})$. Since the values of $L$ and $\Lambda$ shown in Fig. 10 cover most values used in the calculations reviewed in this paper, the sparse sampling scheme guarantees stable numerical routines to get accurate results.

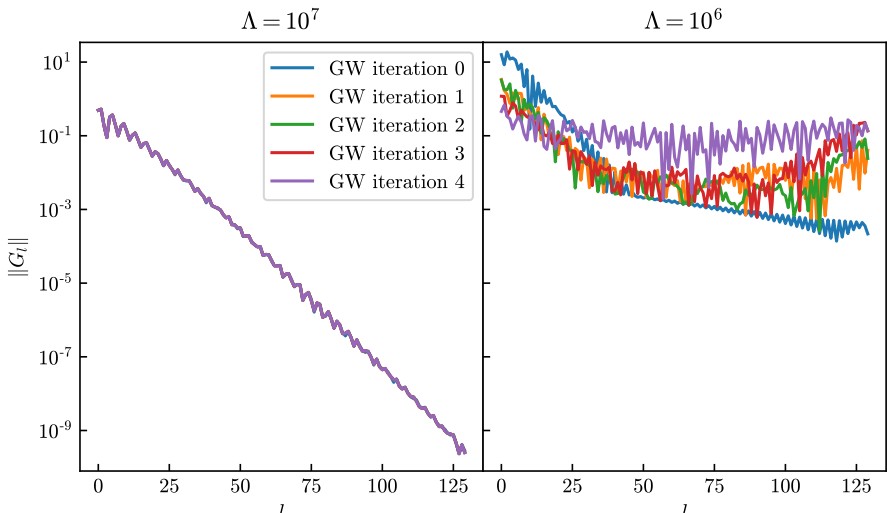

Figure 11: Examples of stable and unstable *GW* calculations of the Krypton atom

In practical applications of the sparse sampling method, one should take care not to introduce large systematic errors in the basis representation as defined in Eq. (9), such as large truncation errors $\epsilon_L$ due to insufficient basis size $L$ or control parameter $\Lambda$. For example, a systematic error at the level of $10^{-3}$ in the basis representation, amplified by a condition number of $10^3$ of the "fitting" procedure, may lead to a numerical error greater than the actual result. As shown in Fig. 11, such a situation could make the simulation unstable. Therefore, to ensure stable numerical calculations with the sparse sampling method, one should choose the appropriate basis parameters $L$ and $\Lambda$ such that the basis representation stays accurate.

Figure 11 shows examples of stable and unstable *GW* calculation of the Krypton atom in cc-pVDZ basis with $\beta = 10\,000\,E_{\mathrm{h}}^{-1}$ using sparse sampling. We carried out five *GW* iterations with different choices of $\Lambda$, and plotted the norms of $G_l$ from each iteration. The left panel of Fig. 11 shows results for $\Lambda = 10^7$. At each iteration, the Green's function is well approximated by the IR basis and the basis truncation error is small, resulting in a stable simulation. In the right panel, a smaller $\Lambda = 10^6$ is used, which is insufficient for this system and introduces a large systematic error (around $10^{-3}$), while the condition number of $\mathbf{F}$ is $\sim 10^3$. The systematic error is amplified by fitting procedure, rendering the *GW* simulation unstable.

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
