# Peer review of "Efficient ab initio many-body calculations based on sparse modeling of Matsubara Green's function"

_SciPost Physics Lecture Notes, doi:SciPost Phys. Lect. Notes 63 (2022)_

## Round 1 · Referee Report · Anonymous (Referee 1) · 2021-10-2

Report

The authors propose a sparse representation method for Matsubara Green's functions that can be used for a reduction in terms of compute time and memory usage when used for DMFT, FLEX, GW and other many-body perturbation theory calculations.

The corresponding time and frequency grids are derived from sampling the extrema of the singular vector function of the Lehman kernel corresponding to the smallest singular value. The weights of the resulting representation are
obtained from a least-square fit (Eqs. 13-14). From a mathematical point of
view, using the infinity norm has typically better compression properties than
the L2 one (see for instance Takatsuka et. al. in JCP 129, 044112 (2008)). The
authors should mention this fact in the text.

Apart from this fact, I find the manuscript concise and ready for publication.

An analytic form of U and V might be obtained as follows. Write Eq. (33) as
an integral and find the corresponding minimax quadrature. This might reveal
the analytic form of the singular functions.

Requested changes

Authors should mention that L-infinity norm outperforms the used L2 in the text.

  • validity: -
  • significance: -
  • originality: -
  • clarity: -
  • formatting: -
  • grammar: -

Author:  Hiroshi Shinaoka  on 2022-06-10  [id 2573]

(in reply to Report 2 on 2021-10-02)

We thank the Referee for the very positive assessment of the manuscript. We are sorry for the slow response, which was due to the release of a new Python library, sparse-ir. We have updated the manuscript according to Comment #1 as described below.

Comment #1

The corresponding time and frequency grids are derived from sampling the extrema of the singular vector function of the Lehman kernel corresponding to the smallest singular value. The weights of the resulting representation are obtained from a least-square fit (Eqs. 13-14). From a mathematical point of view, using the infinity norm has typically better compression properties than the L2 one (see for instance Takatsuka et. al. in JCP 129, 044112 (2008)). The authors should mention this fact in the text.

We thank the Referee for letting us know the relevant paper. We have added the following comment as a footnote above Eq. (17):

Note that the $L_2$ norm is not only the choice in the fitting. It is claimed that using the infinite norm leads to smaller errors~\cite{doi:10.1063/1.2958921}.

Comment #2

An analytic form of U and V might be obtained as follows. Write Eq. (33) as an integral and find the corresponding minimax quadrature. This might reveal the analytic form of the singular functions.

We thank the Referee for the valuable suggestion for future studies. Our attempt to find the analytic form of the basis functions has not been successful.

---

## Round 1 · Referee Report · Anonymous (Referee 2) · 2021-10-3

Strengths

  • review of recent advances in a pedagogical manner
  • open-source implementation
  • several examples for a newcomer

Weaknesses

Not much - some typos and connections with other works as described below.

Report

These lecture notes review recent advances how to define and use an efficient representation for
Matsubara green functions and provide an associated numerical library. I found the presentation pedagogical and
could easily follow it without being an expert on the topic. The numerical library [at least the Python version] is intuitive,
easy to use, and easily embedded into already developed workflows. This will be a valuable material for researchers who want to use these numerical tools.
I will recommend the paper to be published but ask the authors to consider suggestions. I'd strongly recommend authors to reread the whole text carefully, as I found many typos which should not be there is a paper with 9 authors.

Requested changes

  1. Authors made a nice overview of the recent intermediate representation progress, but a further search shows that they missed a recent alternative approach in https://arxiv.org/abs/2107.13094 . Reading through the paper, my impression is that it gives an alternative but mostly comparable approach. I believe it would be useful to include a short discussion on the comparison between the two approaches, when to use one or another, etc.

  2. Do you understand how strict is the condition that a number of IR basis functions grow logarithmically. You showed two examples, but are there known counterexamples. Is there some physical reason for it?

  3. It would be useful to gather weak points of the method in a paragraph of the main text and at which steps should an inexperienced be careful. For instance, the sparse sampling and its numerical stability, can user create precomputed coefficients if they are needed for some other values of $\Lambda \beta$, etc. Are these the only problems?

  4. Typos:

  5. Eq.2: missing \alpha on G
  6. Eq.3: you mark Matsubara propagators with a hat. Why G(\omega) has a hat? I thought its real frequency.
  7. below Eq.5: "while right singular functions [V_0(\omega)]" should probably be functions of omega, right?
  8. what is \epsilon_L in Eq.6
  9. We call these “appropriate frequencies” “sampling frequencies.” Missing or?
  10. Eq. 23: what is the .I at the end?
  11. it is difficult to calculate Eq. Using the IR basis,
  12. . interested readers are

  • validity: top
  • significance: good
  • originality: high
  • clarity: top
  • formatting: good
  • grammar: reasonable

Author:  Hiroshi Shinaoka  on 2022-06-10  [id 2572]

(in reply to Report 1 on 2021-10-03)
Category:
answer to question

We thank the Referee for the very positive assessment of the manuscript. We are sorry for the slow response, which was due to the release of a new Python library, sparse-ir. We have revised the manuscript according to the comments and fixed typos throughout the manuscript. We have updated the example source codes accordingly.

Comment #1

Authors made a nice overview of the recent intermediate representation progress, but a further search shows that they missed a recent alternative approach in https://arxiv.org/abs/2107.13094 . Reading through the paper, my impression is that it gives an alternative but mostly comparable approach. I believe it would be useful to include a short discussion on the comparison between the two approaches, when to use one or another, etc. We thank the Referee for the valuable suggestion. That preprint appeared after we uploaded our manuscript. They proposed a related approach based on a similar decomposition of the same kernel though its accuracy, numerical stability, and efficiency have been established in ab initio calculations. Following the suggestion, we compare the two approaches in Sections 3.4 and 3.5 of the revised manuscript.

Comment #2

Do you understand how strict is the condition that a number of IR basis functions grow logarithmically. You showed two examples, but are there known counterexamples. Is there some physical reason for it? We thank the Referee for asking an important question. As long as $\rho(\omega)$ is bounded, the number of basis functions grows only logarithmically. This is based on the fact that the number of singular values above a certain cut-off value glows only logarithmically [see Fig. 4 of N. Chikano, J. Otsuki, H. Shinaoka, PRB 98, 035104 (2018)]. Although this property/scaling has been verified only numerically, we have not found any counterexamples.

Comment #3

It would be useful to gather weak points of the method in a paragraph of the main text and at which steps should an inexperienced be careful. For instance, the sparse sampling and its numerical stability, can user create precomputed coefficients if they are needed for some other values of $\Lambda\beta$, etc. Are these the only problems? We thank the Referee for the useful suggestion. The choice of $\omega_\mathrm{max}$ is only a major problem in practical calculations. The user should always check if expansion coefficients $G_l$ of each object decay as fast as the singular values to some noise level. If not, the user should increase $\omega_\mathrm{max}$. We have recently developed an updated version of the python library (renamed as sparse-ir from irbasis). The new library allows us to compute IR basis functions for an arbitrary value of $\omega_\mathrm{max}$ on the fly.

We have extended Sec. 3.3 to explain how to check if $\omega_\mathrm{max}$ is large enough and how to tune the parameter otherwise. The sample codes in Section 5 have been adapted for the new python library.

Comment #4

We thank the Referee for pointing out many typos. All the referees have made a critical reading of the manuscript and have fixed typos throughout the manuscript. Section 2 has been revised for better consistency with the new Python library.

---

## Round 2 · Referee Report · Anonymous (Referee 2) · 2022-6-23

Strengths

  1. Authors have updated the text with a comparison to related approaches
  2. Authors have upgraded the numerical library described in the manuscript

Weaknesses

Nothing in particular

Report

I think the authors have made a proper job in the resubmission and I can recommend the paper for publication.

---

## Round 2 · Author Response

Dear the editors,

Thank you for forwarding the all-in-all very positive referee reports with constructive suggestions. Although we were asked for only a minor revision, we unexpectedly decided to upgrade the Python library described in the manuscript. We apologize for the late response, as we had to wait for that.

We answer in detail all points raised by the referees.
We summarize the major updates in the list of changes below.
We believe our manuscript has largely profited from the referees' suggestions
and that the revised version is now suitable for publication.

Yours sincerely,
Hiroshi Shinaoka, Naoya Chikano, Emanuel Gull, Jia Li, Takuya Nomoto, Junya Otsuki, Markus Wallerberger, TIanchun Wang, Kazuyoshi Yoshimi

---

## Round 2 · List of Changes

• The notation in Section 2 has been changed for consistency with the new Python library ``sparse-ir'' The kernel is now defined in imaginary time rather than Matsubara frequency.
  • For a comparison with related approaches, we have extended Sec. 3.4 and added a new subsection 3.5 (comment #1 of the first Referee). A new figure (Fig. 6) has been added.
  • The description of the Python library and sample codes have been updated for the new Python library ``sparse-ir''.
  • Minor revisions to improve the readability have been made.

---

## Editorial Decision

published